# Climate Change and Consumer’s Attitude toward Insect Food

**DOI:** 10.3390/ijerph16091606

**Published:** 2019-05-08

**Authors:** Hsiao-Ping Chang, Chun-Chieh Ma, Han-Shen Chen

**Affiliations:** 1Department of Health Diet and Industry Management, Chung Shan Medical University, No. 110, Sec. 1, Jianguo N. Rd., Taichung City 40201, Taiwan; pamela22@csmu.edu.tw; 2Department of Medical Management, Chung Shan Medical University Hospital, No. 110, Sec. 1, Jianguo N. Rd., Taichung City 40201, Taiwan; 3Department of Public Administration and Management, National University of Tainan, Taiwan No. 33, Sec. 2, Shu-Lin St., Tainan 70005, Taiwan; ccma@mail.nutn.edu.tw

**Keywords:** vulnerability of food systems, food neophobia, environmental concern, global environmental change, behavior change

## Abstract

Given the influence of rising environmental awareness, food systems and security are receiving increasing international attention. Previous studies have discussed the acceptance of insect foods but have been primarily conducted in a European context. Hence, their results cannot be applied to Taiwanese consumers. Regarding this, our study is centered on the theory of planned behavior and considers environmental concern and food neophobia to discuss the effects of consumer attitudes, subjective norms, and perceived behavioral control on the purchase intention toward insect food. We used purposive sampling to survey questionnaire answers face-to-face in Taichung city, Taiwan. We distributed 408 surveys of which 77.45% were used in this study. The results revealed that consumer attitudes, perceived behavioral control, and food neophobia significantly influence purchase intention, whereas subjective norms and environmental concern did not demonstrate significant relationships with purchase intention. According to these results, we suggest that businesses emphasize consumers’ product experience or reduce levels of food neophobia to increase consumer interest in insect foods and improve the acceptability of such foods, thereby increasing purchase intention.

## 1. Introduction

According to a report by the Food and Agriculture Organization (FAO) [1] of the United Nations, the global population will reach 9 billion by 2050. Furthermore, in the light of the myriad of environmental concerns we are faced with, including global climate change, shortages in global arable land, and so on, in response to this rapid population growth, food volumes must increase. Comparing edible insects with average traditional livestock, raising 1 kg of edible insects on average only requires 2 kg of feed, whereas raising 1 kg of beef requires 8 kg of feed. Furthermore, they also help to reduce environmental burdens, emit far less greenhouse gases, and require much less land and water than mammals and birds [2]. The main reason insects can be used as meat substitutes is that they can be used at lower economic and environmental costs. The amount of land used per kg of insect protein is 50–90% lower than that of traditional livestock, 40–80% less per kilogram of edible food, and their GHGEs (greenhouse gas emissions) are 1000–2700 grams lower. The study confirmed that there are five edible insects (*Tenebrio molitor*, *Acheta domesticus*, *Locusta migratoria*, *Pachnoda marginata*, and *Lapatia dubia*) with greenhouse gas and ammonia emissions equivalent to pigs, but far less than the emissions of cattle. In addition, the amount of carbon dioxide produced per kilogram is lower than that of traditional livestock [3].

Past literature has evidenced that insects have important effects in the nutritional history of Africa, Asia, and Latin America [4,5,6], such as the Yukpa people in Colombia and Venezuela who use certain insects in their meat dishes. There are also studies exploring *zonocerus variegatus*, bee larvae, and pupae in southern Nigeria, which are all rich in protein [5,7]. Researchers have shown that insects are a good source of protein and micronutrients [5,6,8]. Zielińska, Baraniak, Karaś, Rybczyńska, and Jakubczyk [9] pointed out that insects have high levels of monounsaturated fatty acids and polyunsaturated fatty acids, which satisfy human requirements for amino acids. Moreover, Poma, Cuykx, Amato, Calaprice, Focant, and Covaci [10] noted that compared to common animal products (meats, fish, eggs), insects (locusts) can even serve as a replacement for common proteins. That study encouraged the consumption of insect foods. Existing literature refers to insect foods as insect-based food [11], insect food [12], and insects as food [13]. This indicates that insect foods do not yet have a standard name, although a global population of 2 billion people considers insects a part of their traditional diet and more than 1900 types of insects are used by humans for food [1]. In remote rural and biodiversity-rich tropical countries, insects have been an important source of protein and micronutrients for thousands of years [6], with the world’s most common edible insects accounting for 31% of beetles, followed by caterpillars, bees, wasps, ants, etc. [2]. In Europe, edible insect foods are relatively novel foods [8], and companies already sell edible insect products, such as the French company Micronutris, which sells the cricket *sigillatus*, tablette-tenebrio’s chocolate, aldente-aenebrio pasta and crackers, etc., [14] while Finnish chain bakery Fazer sells ‘Fazer Cricket Bread’ made from glutinous flour [15]. Not only that, the European Union’s non-profit organization, IPIFF (International Platform for Insects as Food and Feed, IPIFF), promotes the use of insects and insect-derived products as a source of human consumption and animal feed, and believes that it can be cultured with fewer resources (such as land, water, feed, energy), while generating lower greenhouse gas emissions and pollutants. It is likely to be the most common source of protein for aquaculture and livestock animals, and it is believed that insect proteins will become a generally accepted dietary component of Western society. Not only Western countries however, as palm weevil and cricket have also been cultivated on a commercial scale in Thailand [16], and similar commercial farming models are being developed in Africa [17]. In Taiwan, although it started late, there is an ecological rehabilitation of the Fuyang enterprises that have both green energy and alternative food. In Europe, foods such as insect flour and insect crackers exist. In Asia, Thailand has a well-developed insect-eating culture, and insect dishes including fried silkworm pupae and fried crickets are eaten in Taiwan. This study’s discussion on this topic means that insect foods, as food products which have insect ingredients, are on the market and can be presented in any form. For example: cricket biscuits, cricket bread, fried insects (such as grasshoppers, pupae, meal-worms) and so on.

Tan, Tibboel, and Stieger [18] demonstrated that when a novel food, such as insects, is mixed with a familiar product, this can generate positive expectations and improve the sensory experience and attractiveness of the novel food, thereby increasing consumer approval and helping reduce barriers to consumption. In Hartmann, Shi, Giusto, and Siegrist’s [19] study comparing the cultures of Germany and China regarding insect eating, Chinese people were found to evaluate insect foods according to their taste, nutritional value, familiarity, and social acceptance. That study indicates that if insects could be combined with a familiar food, this could possibly encourage their use as a source of food in the West and reduce consumers’ negative attitudes toward insect foods. Research related to consumers and edible insect food products is not very common, especially in Taiwan, but what does exist includes the manufacturing methods of products [20], consumer knowledge [12], and food suitability [20].

Numerous studies have examined the link between insect foods and purchase intention [12,21,22]. To predict and explain human behaviors, social psychologists have proposed several models. The most frequently utilized among these is the theory of planned behavior (TPB), which is commonly considered as being effective in predicting general behavior. The TPB argues that the following three factors, in combination, are responsible for forming behavioral intentions—perceived behavioral control, attitude toward the behavior in question, and subjective norms [23]. Based on the theory of planned behavior, Ajzen [24] pointed out that perceived behavioral control means “an individual’s perceived ease or difficulty in performing a particular behavior”. An attitude toward a behavior is considered to be the “degree to which a person has a favorable or unfavorable evaluation or appraisal of the behavior in question”. “Subjective norms” can also be divided into individual behavioral norms and social norms. Personal behavioral norms mainly come from the influence of important reference objects formed by parents, friends, peers, or experts. Social norms refer to the influence of pressure from other social groups to perform or not perform the behavior. This means that an individual who possesses a reasonably high level of behavioral control for a particular behavior will have a higher chance of show-casing a firm intention to engage in that behavior. “Behavioral intention” indicates a person’s willingness to conduct a certain behavior, with the assumption that this willingness must necessarily exist before the behavior itself is executed [21]. Throughout its development, the TPB has been widely applied by scholars to study purchase intention [25,26,27,28,29]. Menozzi et al. [30] used the TPB to investigate the intentions and expected behaviors of young Italian consumers on the consumption of flour and chocolate biscuits containing 10% crickets. Their study found that the primary obstacles to the consumption of insect-containing flour products is the aversion generated from seeing nearby insects, nonconformance to local food culture, and the lack of related products in supermarkets. According to Menozzi et al. [30] in relation to insect foods, modern consumers maintain a certain degree of reservations toward purchasing insect food products. This may be the result of consumers’ aversion to insects, incompatible local food cultures, or consumers not having seen such products before.

Previous studies have noted that when compared with familiar foods, a consumer’s willingness to choose new foods, or those that have not been previously eaten, may be affected by neophobia [31,32,33]. Pliner and Hobden [31] referred to food neophobia as the fear of food that has either not been eaten or seen in an eating situation, and describe an unwillingness to approach and eat unfamiliar food. People generate a mixture of conflicting curiosity and fear toward foods they have not previously seen [34]. Pliner and Hobden [31] developed the food neophobia scale (FNS) to measure aversion to new foods, and previous studies have utilized the FNS to evaluate consumer willingness to eat or choose new foods [18,31,35,36], familiarity and experience with foreign foods [31,33], willingness to explore food flavors [35,37], and expectations from new foods [31,36]. Some scholars have applied a food technology neophobia scale to measure consumer attitudes toward new and traditional food technologies [38], university students’ satisfaction regarding food-related life [39], and consumer neophobia to red wine [40]. In addition, the degree of an individual’s food neophobia influences their purchase of food products and frequency of purchase [41], as well as their eating habits [42]. Along the same lines, La Barbera et al. [21] noted that consumer food neophobia and aversions affect the willingness to eat insect foods. The present study postulates that due to varying levels of food neophobia among Taiwanese consumers, consumers will have varying reactions, which may influence their willingness to purchase insect food products. For these reasons, the present study includes food neophobia as a research variable.

Whether consumers are willing to purchase a product does not depend simply on their preference for the product. Rather, environmental awareness tactics influence consumer behaviors, such as emphasizing low carbon foods and beverages, encouraging consumers to support local foods, supporting the purchase of in-season, organic, and fair-trade products, and focusing on product labeling [43,44]. Junior et al. [45] observed that individual views on environmental concern affect ones willingness to purchase green products. As a result, this study uses environmental concern as a research variable to study whether consumer willingness to purchase insect foods is influenced by environmental concern.

In summary, this study uses the TPB as a theoretical core for investigating the effects of consumer attitudes, subjective norms, and perceived control on the willingness to purchase insect foods. It also adds the two elements of environmental concern and food neophobia. Through this, the study aims to provide an understanding of the likelihood of insects being used as food in Taiwan. In the event that the insect food-product industry develops, these results can serve as a reference for the development of marketing strategies.

## 2. Materials and Methods

### 2.1. Research Framework

This study is centered on the TPB and includes two variables—environmental concern and food neophobia. It investigates Taiwanese consumer knowledge of, and attitudes toward, insect foods, and forecasts consumer purchase intention. The research framework of this study is shown in Figure 1.

### 2.2. Literature Review

#### 2.2.1. Purchase Intention

Purchase intention is defined as the likelihood of a consumer purchasing a given product. A higher purchase intention indicates an increase in purchase probability [46,47,48,49,50]. Marketers have long argued that purchase intention can be used to accurately forecast purchase behavior [51,52,53]. Moreover, previous studies have explored product knowledge [54,55,56,57], food neophobia [41,58,59,60], and purchase intention. In this study we define the definition of purchase intention as a customer’s intention to purchase insect food.

#### 2.2.2. Attitude

Ajzen [23] believes that attitude is a product of an individual’s behavioral beliefs and outcome evaluations [61]. Studies have shown that consumer attitudes toward organic foods [62], green foods [63], and green products [26,64,65,66] affect purchase intention. Furthermore, Asif, Xuhui, Nasiri, and Ayyub [67] observed that attitude and health consciousness may more accurately forecast purchase intention toward organic products. As such, this study postulates that consumer attitudes toward insect foods will influence their purchase decisions. On this basis, we propose hypothesis 1:
**Hypothesis 1 (H1).** Consumer attitude has a significant and positive influence on purchase intention toward insect foods.

#### 2.2.3. Subjective Norms

Ajzen [23] argued that subjective norms are social pressures (e.g., opinions of relatives, close friends, or colleagues) that influence decision making and whether to take action [68,69]. Additionally, Scalco et al.’s [62] study showed that subjective norms significantly influence purchase intention toward organic foods. On the other hand, this study speculates that consumers are influenced by their relatives and friends when making decisions regarding whether to purchase insect foods. On this basis, hypothesis 2 is proposed:
**Hypothesis 2 (H2).** Consumers’ subjective norms positively and significantly influence their purchase intention toward insect foods.

#### 2.2.4. Perceived Behavioral Control

Ajzen [24] argued that perceived behavioral control is the ease or difficulty with which individuals perceive the accomplishment of a certain behavior. This control is affected by external factors, which influence these behaviors, meaning that individuals may be subject to obstacles from past experiences and expectations, including their understanding of their own abilities (ability), urgent needs (resources), and convenience (opportunity). Yadav and Pathak [70] observed that perceived behavioral control significantly influences consumers’ purchase intentions with regard to organic foods. With this, we propose hypothesis 3:
**Hypothesis 3 (H3).** Consumers’ perceived behavioral control positively and significantly influences their purchase intention toward insect foods.

#### 2.2.5. Environmental Concern

Minton and Rose [71] define environmental concern as the general attitude toward preserving the environment. Newton et al. [72] assert that environmental concern does not directly affect purchase intention but rather allows consumers to understand the environmental result produced by purchasing a product. Furthermore, Arisal and Atalar [73] note that collectivists are more attentive toward environmental topics and find that environmental concern affects individual purchase intention. Collectivists involve emotions such as empathy and indebtedness [74]. Because of their focus on society rather than the individual, public benefits are often considered altruistic [75,76]. Some examples of public benefits include improved environmental outcomes and enhanced animal welfare [77]. The new ecological paradigm measures individual perspectives toward environmental attitudes [78,79,80]. Lee [81] notes that environmental concern is the second major index for forecasting green purchasing behaviors. In summary, environmental concern is correlated with purchase intention, and on this basis, we propose hypothesis 4:
**Hypothesis 4 (H4).** Consumers’ environmental concerns positively and significantly affect their purchase intention toward insect foods.

#### 2.2.6. Food Neophobia

Studies have shown that individuals with high degrees of food neophobia purchase some types of foods (e.g., poultry and fish) at a lower frequency and others (e.g., pork) at a higher frequency [41]. A study by Jaeger et al. [42] confirms that the eating habits of adult New Zealanders are subject to the varying influences of different degrees of food neophobia. Adults with high degrees of food neophobia have lower intake frequencies and selection norms for foods such as fruits, vegetables, protein, seasonings, beverages, and dairy products when compared to those with low degrees of food neophobia. According to this analysis and the definitions and topics of this study, food neophobia is defined as the degree to which consumers are unwilling to consume insect foods when they experience dislike or fear [31,32]. This study also uses the FNS to measure the degree of food neophobia, which reveals that when consumers have higher degrees of food neophobia, stronger negative emotions result, refusals to eat certain foods become more intense, and purchase intention decreases. Previous studies have also stated that in some Western countries, people’s dislike toward and refusal to eat insects is a kind of food neophobia [30]. As such, this study argues that because insect foods are a relatively unfamiliar food product for Taiwanese consumers, they may experience fear or disgust when they see insect foods. On this basis, we present hypothesis 5:
**Hypothesis 5 (H5).** Consumers’ food neophobia negatively and significantly affects their purchase intention toward insect foods.

### 2.3. Questionnaire Design

The questionnaire design of this study comprises six sections and is presented in Appendix A. Sections one to three deal with the scale for the TPB. In these sections, questions related to attitude and subjective norms refer to the study by Menozzi et al. [30] and investigate the perception of consumers toward the consumption of insect food products using three questions, and whether consumers are influenced by reference objects and purchase insect food as a result using another three questions. Next, the questionnaire addresses perceived behavioral control, referring to the studies by Ajzen [23] and Menozzi et al. [30] to investigate the ease with which consumers purchase insect foods. This section includes four items, with a total of ten questions. The fourth section addresses consumers’ environmental concern, referring to the research by Dunlap and Van Liere [78] to establish a scale for environmental concern and investigate consumer perceptions and attitudes toward the environment. The fifth section addresses consumer food neophobia with regard to insect foods and applies a scale developed by consulting the work of Pliner and Hobden [31], Siegrist et al. [41], and Jaeger et al. [42]. It includes six questions that measure respondents’ individual levels of food neophobia. The sixth section addresses consumer willingness to purchase insect foods. The studies by Chen and Cheng [82], Chen et al. [83], and Singh and Verma [84] were consulted to develop a scale for purchase intention and determine the likelihood that consumers will purchase insect foods. In lists 1–6 (Appendix A), a 7-point Likert scale is used, allowing respondents to provide ratings ranging from “strongly disagree” (1) to “strongly agree” (7). The seventh section addresses basic information about respondents, including their gender, profession, education level, and average monthly income.

### 2.4. Sample Size and Composition

In this study, we used purposive sampling to survey questionnaire answers face-to-face in Taichung city, Taiwan. We distributed 408 survey forms and 100% of these were returned. After eliminating 92 invalid survey forms, 316 valid survey forms remained for a valid return rate of 77.45%. Regarding the respondents’ demographics, slightly fewer were women than men (41.1% versus 58.9%). The largest age group among the respondents were those aged 31–40 years (38.4%), followed by 41–50 years (31.5%). The most common highest level of educational achievement was senior high school (42.5%), followed by university (30.6%). Over half of the respondents worked in the service industry (38.6%) or the traditional manufacturing industry (22.8%). The average monthly income among the respondents was most commonly between NT$40,001 and NT$50,000 (47.3%), followed by between NT$30,001 and NT$40,000 (22.9%).

### 2.5. Statistical Analysis

This study adopted structural equation modeling (SEM) to examine the structural relationship between attitude, subjective norm, perceived behavioral control, environmental concern, food neophobia and purchase intention, and model fit. The SEM is an effective model test and improvement method that enables theoretical models to be tested and can explain the causal relationships among the variables in hypotheses which are related to the models based on statistical dependence. The analysis used the SPSS version 21.0 statistical software package (IBM Corp.: New York, NY, USA) and Amos version 21.0 (IBM Corp.: New York, NY, USA).

## 3. Results

### 3.1. Measurement Model: Reliability and Validity

Reliability and convergent validity analysis results for each construct are presented in Table 1. Fornell and Larcker [85] stated that a Cronbach’s α coefficient of greater than 0.7 indicates high reliability, whereas a coefficient lower than 0.35 indicates low reliability. The Cronbach’s alpha of each dimension was greater than 0.80, indicating good reliability [86]. Hair et al. [87] suggested that the composite reliability (CR) of latent variables should be greater than 0.70. A high CR of latent variables for an examined variable indicates that the examined variable is valid for use in measuring the latent variable. The CR of the variables in this study ranged from 0.713 to 0.940, indicating that this model had good internal consistency. The average variance extracted (AVE) for each factor was between 0.560 and 0.839, which is higher than the recommended benchmark of 0.5 [85]. However, the value of factor loading (0.624–0.952) was higher than the recommended level of 0.6 [88]. Means, standard deviations, and correlations among the constructs are presented in Table 2. Significant positive correlations were found to exist between attitude and purchase intention (*r* = 0.39, *p* < 0.01), subjective norm and purchase intention (*r* = 0.40, *p* < 0.01), perceived behavioral control and purchase intention (*r* = 0.31, *p* < 0.01), and environmental concern and purchase intention (*r* = 0.27, *p* < 0.01). These results indicate that the higher the attitude, subjective norm, perceived behavioral control, and environmental concern of consumers, the stronger their purchase intention to insect food. By contrast, food neophobia exhibited a significant negative correlation with purchase intention (*r* = −0.40, *p* < 0.01), indicating that the higher the food neophobia of consumers, the weaker their purchase intention to insect food.

### 3.2. Structural Model: Goodness-Of-Fit Statistics and Hypothesis Testing

AMOS version 21.0 was first used to conduct confirmatory factor analysis (CFA). Five latent constructs were contained within the measurement model (Figure 1). As shown in Table 3, the revised model exhibited an appropriate fit after CFA (χ^2^/df = 2.174, goodness-of-fit index = 0.908, root mean square error of approximation = 0.069, comparative fit index = 0.932, normalized fit index = 0.946, adjusted goodness-of-fit index = 0.873).

## 4. Discussion

SEM was utilized in the research, and a maximum likelihood estimation was employed to measure the associations among attitudes, subjective norms, perceived behavioral control, environmental concern, food neophobia, and purchase intention. Figure 2 demonstrates standardized path coefficients arising from examining the proposed structural model. The results of this study reveal that consumer attitudes significantly and positively affect purchase intention toward insect foods (β = 0.772, *p* < 0.001), thus supporting H1. This also indicates that when consumers believe that an insect food tastes good or is acceptable, their attitudes toward insect foods are more positive and they are more likely to purchase them. As such, consumer attitudes do influence purchase intention and this result is consistent with previous literature. For example, Yazdanpanah and Forouzani [25] stated that student attitudes toward organic foods are an important factor for predicting purchase intention. Another study showed that attitudes toward the use of smartwatches not only strongly influence willingness to use such products, but also strengthen the purchase intention of potential consumers [89].

However, the results of this study found that consumers’ subjective norms toward insect foods are not significantly correlated with purchase intention (β = 0.062, *p* > 0.05) and therefore, H2 is not supported. This indicates that consumers will not purchase insect food as a result of promotions by famous people or environmental groups. The results of this study are consistent with those of Tan, Ooi, and Goh [90], who state that consumers’ subjective norms related to energy-saving home appliances are not significantly correlated with purchase intention. Yadav and Pathak [70] nonetheless observed that young consumers’ willingness to purchase green products can be predicted through subjective norms. This study postulates that this is probably because insect foods are not consistent with Taiwanese food culture, and therefore consumers are not likely to try such foods on a whim. With regard to home appliances, it is likely that the variation in the previous research results is because there are few brands that are respected by consumers, meaning consumers do not consult the opinions of others on this topic.

Consumers’ perceived behavioral control is positively and significantly correlated with purchase intention (β = 0.564, *p* < 0.05), thus supporting H3. This indicates that when a consumer believes they can easily obtain insect foods, their purchase intention will increase. According to TPB, perceived behavioral control influences customers’ intention and behavior because customers use perceived behavioral control to judge the likelihood of successfully performing the behavior or accomplishing the task [91]. Perceived control can increase as a result of a convenience channel to use, a channel which the customer is comfortable with or has experience using, or a channel which provides clear and transparent information about the process from purchase to delivery. If a channel provides a customer higher perceived control, then their purchase intention also increases [92]. On the other hand, the perceived difficulty of using a channel can result from obstacles and challenges in searching for products or interacting with the vendor, which can easily prevent the customer from making a purchase [93]. The results of this study are the same as those of previous literature, which state that consumers’ perceived behavioral control with regard to green skincare products positively and significantly affects their purchase intention [94]. Perceived behavioral control is also an important factor in determining consumers’ intentions to purchase local pork products [95]. In other words, accessibility is an important factor related to purchasing products. With regard to environmental concern, this study reveals that it is not significantly correlated with purchase intention (β = 0.075, *p* > 0.05) and therefore, H4 is not supported. This indicates that consumers do not purchase insect foods as a result of environmental influences. The results of this study are consistent with those of Asif et al. [67], who conclude that environmental concern does not significantly influence consumers’ intentions to purchase organic food products. However, Prakash and Pathak [96] state that consumers’ environmental concern positively and significantly affects their intention to purchase products with environment-friendly packaging. This study postulates that although insect foods, organic foods, and products with environment-friendly packaging are helpful for the environment, conceptually, consumers do not consider insect foods as environment-friendly, and therefore environmental concern is not a factor influencing purchase intention. Furthermore, with regard to consumer perception of organic foods, consumers likely view such products from a health perspective and therefore, environmental concern is not included as an influencing factor.

Finally, consumers’ food neophobia was significantly and negatively correlated with their intention to purchase insect foods (β = −0.526, *p* < 0.001), thus supporting H5. This indicates that when consumers have higher degrees of food neophobia with regard to insect foods, they will not purchase them. This is consistent with the results of Piha et al. [12], La Barbera et al. [21], and Tan et al. [22], which state that consumers with high degrees of food neophobia are unwilling to try new functional foods [97] and this, in turn, reduces purchase intention. But Piha et al. [12] suggested that differences in cultural regions also affect consumers’ purchasing intention to buy insect food. When consumers know more about edible insect food, they will have a higher willingness to buy it. Food neophobia is also correlated to consumer interest in food [41] and can affect consumers’ hedonic assessment of food. Jaeger et al. [42] identified food neophobia as the consumer preference for certain kinds of food, and an important impediment to changing food habits and resolving food-related health problems. As such, resolving consumer food neophobia toward certain types of foods is helpful for improving consumer acceptance of such foods. Per these findings, H1, H3, and H5 are supported but H2 and H4 are not. A summary of the verification of the hypotheses made in this study is shown in Table 4.

## 5. Conclusions and Limitations

### 5.1. Conclusions

The results of this study reveal that for consumers, attitude, perceived behavioral control, and food neophobia have significant effects on their purchase intention. Scholars have identified that when consumers have positive feelings about their experiences, their attitudes toward products are more positive [22,98,99]. Therefore, this study shows that edible insect companies can enhance consumer knowledge and experience of edible insect products. For foods that consumers are not familiar with or that they have never eaten before, or edible insect foods that they are afraid of trying, edible insect companies can educate consumers about edible insects, as well as edible insects with high protein and nutrients. Consumers can also reduce fear or uncertainty about eating edible insect products through product experience. Furthermore, edible insects can be considered as sustainable food because insects are simply far less demanding than beef, e.g., 7 times less feed, 50 times less water, and 100 times fewer greenhouse gases [100]. The cultivation of edible insects can reduce the environmental burden and contribute to environmental protection more than the conservation of traditional livestock [101]. In addition, edible insect products are also sold in other countries. For example, the French company Micronutris sells the cricket *sigillatus*, tablette-tenebrio’s chocolate, and aldente-aenebrio pasta and crackers [102], and the Fazer chain bakery are going to start selling ‘insect bread’ and offer a product called Fazer Cricket Bread [103], and so on. Fazer is one of the largest corporations in the Finnish food industry. Edible insect companies can cook insect foods in a familiar cooking style [13] or introduce insect foods into common products [90]. These methods can increase consumer interest and acceptability of insect foods, thereby increasing the intention to purchase.

### 5.2. Limitations of the Research and Future Research

In the context of global appeals for environmental issues, modern consumers are attentive to environmental, ecological, and sustainable development topics. However, insect foods are environmentally friendly and sustainable products, like organic foods and green products, and although relevant magazine articles have reported this fact, this has yet to be widely accepted by consumers. As such, it is recommended that future studies compare consumer attitudes towards different environmentally friendly products (e.g., organic foods and insect foods), including environmental awareness, behaviors, and values, and compare the variation among consumer knowledge. Additionally, it is recommended that future studies analyze the knowledge and perspectives of consumers belonging to various age groups or different cultural backgrounds (e.g., Thai and Indian consumers). One of the study’s main limitations is the use of a purposive sampling technique, which does not allow the findings to be generalized to the overall population of the customers’ attitudes toward insect food in Taiwan. Future research could also investigate the effect of other customer-related constructs, such as living in different regions, between urban and non-urban responses. In addition, the cost of purchasing insect foods is another future research limitation.

## Figures and Tables

**Figure 1 ijerph-16-01606-f001:**
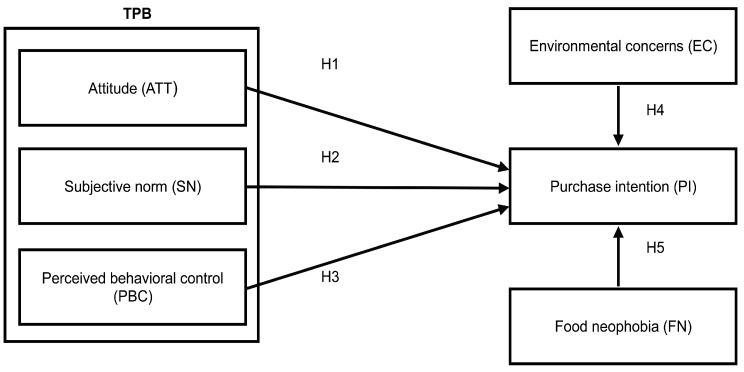
Study hypotheses and conceptual framework. H1, attitude positively affects purchase intention; H2, subjective norms positively affect purchase intention; H3, perceived behavioral control positively affects purchase intention; H4, environmental concern positively affects purchase intention; H5, food neophobia positively affects purchase intention.

**Figure 2 ijerph-16-01606-f002:**
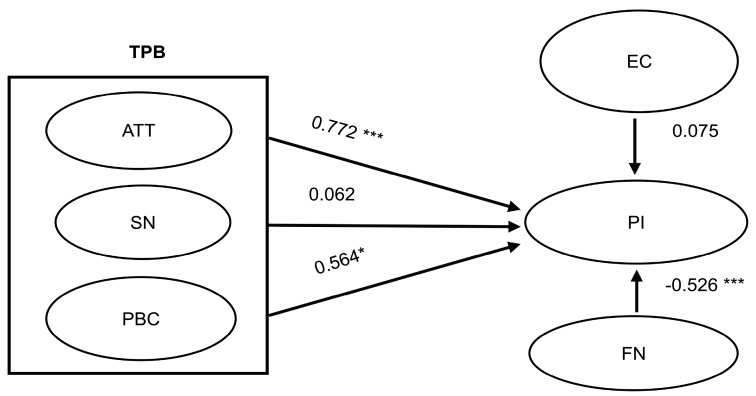
Results of structural equation modeling. * *p* < 0.05; *** *p* < 0.001; GFI = 0.908; AGFI = 0.873; CFI = 0.932; TLI = 0.733; RMSEA = 0.069.

**Table 1 ijerph-16-01606-t001:** Results of factor loading, reliability, and validity.

Items	Factor Loading	Cronbach’s α	AVE	CR
Attitude		0.917	0.795	0.921
Eating insect food is pleasant.	0.952			
Eating insect food is relevant.	0.791			
Eating insect food is tasty.	0.924			
Subjective Norm		0.874	0.778	0.875
I would buy insect food because: doctors/nutritionists are in favor.	0.862			
I would buy insect food because: environmental groups are in favor.	0.902			
Perceived Behavioral Control		0.674	0.560	0.713
It would be very easy for me to buy insect food.	0.855			
It is mostly up to me whether to buy insect food.	0.624			
Environmental Concern		0.898	0.706	0.905
Space and resources on the Earth are limited.	0.902			
Humans must strive for harmonic coexistence with nature for survival.	0.950			
Humans have overly exploited nature.	0.749			
The balance of nature is delicate and fragile.	0.740			
Food Neophobia		0.873	0.710	0.880
I dare to try new and different foods.	0.892			
Insect food looks too strange	0.799			
I like food from different countries.	0.835			
I will try insect food on specific occasions.	0.746			
I will be afraid to eat food that I have never eaten before.	0.725			
I will eat almost all types of food.	0.758			
Purchase Intention		0.939	0.839	0.940
I would consider buying insect food.	0.934			
I am willing to recommend others to buy insect food.	0.906			
I intend to eat insect food in the future.	0.908			

Note: CR: Composite reliability; AVE: Average variance extracted.

**Table 2 ijerph-16-01606-t002:** Means, standard deviations, and correlations of constructs.

Construct	Mean	S.D.	1	2	3	4	5	6
1. Attitude	4.23	0.62	1.00					
2. Subjective Norm	3.17	0.73	0.21 **	1.00				
3. Perceived Behavioral Control	4.06	0.68	0.23 **	0.32 **	1.00			
4. Environmental Concern	5.68	1.26	0.36 **	0.31 **	0.32 **	1.00		
5. Food Neophobia	4.51	0.74	0.47 **	0.38 **	0.30 **	0.37 **	1.00	
6. Purchase Intention	4.84	0.46	0.39 **	0.40 **	0.31 **	0.27 **	−0.40 **	1.00

Note: ** *p* < 0.01.

**Table 3 ijerph-16-01606-t003:** Results of the fit indicators of the evaluation model.

Fit Index	Ideal Value	Result	Conclusion
χ^2^/df	<3	2.174	Acceptable
GFI	>0.9 (good fit)	0.908	Good fit
	0.8–0.89 (acceptable fit)		
AGFI	>0.9 (good fit)	0.873	Acceptable
	0.8–0.89 (acceptable fit)		
NFI	>0.9	0.946	Acceptable
CFI	>0.9	0.932	Acceptable
RMSEA	≤0.05 (close fit)	0.069	Fair fit
	0.05–0.08 (fair fit)		
	0.08–0.10 (mediocre fit)		
	>0.10 (poor fit)		

Note: GFI: goodness-of-fit index; AGFI: adjusted goodness-of-fit index; NFI: normalized fit index; CFI: comparative fit index; RMSEA: root mean square error of approximation.

**Table 4 ijerph-16-01606-t004:** Summary of hypothesis verification.

Hypothesis	Content	Verification
H1	Attitude positively affects insect-food purchase intention	Supported
H2	Subjective norms positively affect insect-food purchase intention	Not supported
H3	Perceived behavioral control positively affects insect-food purchase intention	Supported
H4	Environmental concerns positively affect insect-food purchase intention	Not supported
H5	Food neophobia will negatively affect purchase intention toward insect food	Supported

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
