# Peer review of "Climate Change and Consumer’s Attitude toward Insect Food"

_ijerph, 2019, doi:10.3390/ijerph16091606_

Round 1

Reviewer 1 Report

Overall, I commend the author for contributing much-needed work in this area. This is a relevant and interesting topic of research. Please see my comments below.

Abstract: 77.45% is not the response rate. Please check your wording and definitions. (Later you refer to this number as your collection rate).

Methods: how was the survey distributed? How was your population determined? There are some important missing pieces to your methods description.

Lines 30-31: We, in fact, produce enough food to feed everyone on earth more than sufficiently at the present time. The issue is one of access and distribution, not production. If you claim production is the issue, please provide citations to support this claim, but also note the Food and Agriculture Organization of the United Nations has noted that we produce enough food to feed every person on earth 2,700 calories/day (http://www.fao.org/3/x0262e/x0262e05.htm). I’m not sure that SUPPLY is the main argument for eating insects – as much as other drivers such as climate change, land use, resource use, nutrition, etc. Consider reframing this issue given this information.

Line 36: what do you mean by “additional danger”?

Line 37-38: how insects-as-food is named is not the same as a definition, as you have pointed out. How are each of these names (insect food, insect-based food, insects as food) defined? If they are defined as eating insects, then the definition is not the issue. This is not clear in how you have phrased it. In other words, how do the definitions in these other studies differ from your definition, provided on line 43?

Line 44: You say “novel food” on line 44 – are insects really a novel food, or is this just in a specific context (i.e. Western, global north, Eurocentric diets)

Lines 60-66: you use several quotations here that are not cited with a page number

Line 73: you say “according to previous research” but then do not provide a citation.

Line 125: are you really changing /narrowing the definition of purchase intention, or are you simply applying the defined concept to eating insects?

H1/H3: Why only positively influence? Could this not also negatively influence? Or do you mean that ANY pre-existing ‘attitude’, whether positive or negative, will influence the consumer? This may just require a simple rewording for clarification.

You mention repeatedly the idea of green consumption – or environmental awareness, and yet you have not outlined the ways that insect proteins are related to environmental concerns. This needs to be made explicit prior to assertions of “environmental concern”.

Is it worth mentioning the acceptable limits for insects “parts” in processed foods? How we are ingesting insects regularly with processed and packaged foods? (https://www.ncbi.nlm.nih.gov/books/NBK221564/)

Line 202: I think what you are trying to say is that 77.45% of the responses were included in the study (rather than this being a “response rate” or “collection rate”).

Line 250: New paragraph for H2?

Line 259-261: beginning with “with regard to home appliances…” unnecessary, consider removing.

H3: what if they don’t believe they can easily access the foods? Does this negatively impact their decisions?

Line 272: is this contradictory to what you wrote previously: “Lee [66] notes that environmental concern is the second major index for forecasting green purchasing behaviors.”

Line 277-278: this is a great point – again, as I mentioned earlier, I would like you to provide a better overview of insect food as an environmental product. Provide some background literature on this (for example, impacts on land and water usage, GHG emissions, etc). As this is one of your main hypotheses, providing the background would provide more substance to your argument and provide the reader with a more comprehensive view of the issue.

Figure 2: can you reposition this in the text closer to where you talk about it?

Limitations? You have identified only 2 variables to purchase intention which, of course, does not provide a comprehensive overview of factors influencing purchasing intentions.

Author Response

 Please refer to the attached file (Annex 1)

Reviewer 2 Report

This manuscript describes a very interesting research endeavour exploring drivers and barriers to purchasing insect foods in Taiwan. However, the manuscript itself requires substantial revisions in order to properly present this research.

The research itself is fascinating and appears to based on a sound theoretical background and grounded in appropriate statistical tests. However, the content of the Methods, Results and Discussion section could benefit from being adjusted to more appropriately fit in with standard format of journal articles and content of these sections.

The title of the article references climate change, but the author does not make the link to how eating insect food will interact with climate change. The author mentions several times throughout the manuscript that eating insect food may be an environmental choice, but do not clearly introduce the idea of insect food as a sustainable alternative protein source.  Some additional context in the introduction about the environmental sustainability of entomophagy would be helpful for the reader.

Introduction:

P1 line 36: what is meant by “additional danger”, the author has not introduced any dangers related to consuming other foods. Are there dangers related to consuming insects?

P1 line 42-43: In providing a definition for insect foods, the author states “in any form containing insects”. While I interpret this as the authors referring to foods intentionally containing insects, it’s worth noting that grains (and likely other products) have a threshold acceptable level of insect content (in parts per billion, I believe) because of the inevitable and unintentional introduction of insects during harvesting and processing stages.

P2 line 51-54: The author states that research related to consumers and insect foods is scarce, but that numerous studies are available in the next paragraph. This is confusing.

P2 line 78: Please define what is meant by “neophobia”.

Materials and Methods

The author fails to provide any information about how the questionnaires are administered. Are they administered by mail, online, over the phone, on the street? How was the sample selected? People attending an event, people visiting a specific website?

Where in Taiwan are were the questionnaires administered, in Taipei, in another region, broadly across the country? If it were possible it would be interesting to look at any differences between urban and non-urban responses.

What language(s) was the survey administered in? I would also consider including a copy of the questionnaire (and an English translation, if appropriate) as an appendix/supplemental document.

Section 2.2:

The methods section contains a lot of background information on the research supporting the hypotheses explored by this project. It is important to provide the background and the theoretical frame on which the research questions are built, however, this is better suited in an introduction section. This could be integrated with the information already included in the introduction to give a more comprehensive introduction to the TPB other drivers.

I would recommend most of this content be moved to an introduction section or subsection, and the hypotheses be highlighted here.

P4 line 140: My understanding of subjective norms direct people to act and to not act, in a given scenario. I don’t think it’s accurate to suggest that they will positively influence purchase intention, especially in the current context, where I believe that entomophagy is not the norm.

P 4 line 154: What are collectivists?

Section 2.3

Were any of the survey questions validated? Do we know how closely linked purchase intention, as measured by these questions is to actual purchasing behaviours?

In my local context, insect foods are a novelty and cost is a barrier. Was cost considered as a factor influencing purchase intention at any point in designing this questionnaire?

Section 2s.5: I believe this is a typo and should read 2.5

P5 line 211: What does SEM stand for? Please describe all acronyms at first use.

P5 line 211-213: This is the first time the author has introduced the idea of “health values” and “health orientation” as factors influencing purchase intention. I would recommend either re-framing this in terms of the hypotheses described previously, or expand on what this analysis is looking for.

Results

P5 line 220: What pre-test? Is this a statistical pre-test, or did the authors pilot the questionnaire before administering it. Either way, this should be described in the methods section.

P5 line 225: Were these questions omitted from the questionnaire or from the analysis?

P5 line 226 - 228: What are standard values?

Table 1: Please provide interpretation of the results presented here in Table 1.

Table 2: Please include in the header that the 1, 2, 3…. Columns are presenting correlation results.

Here and in the text, it’s not clear what kind of correlation test was used or how this was tested. Was the author testing correlation of means across the groups? Further detail regarding the statistical testing is required.

P7 line 233: Should this reference be to table 3?

Discussion

Paragraph 1: There are a lot of results presented here that would be better suited in a results section. Further, the model output (beta and p values) are presented, but as I am not very familiar with SEM, I’m not sure how to interpet the beta values. Can it be expressed in a way that suggests the strength of the association between the model construct and purchase intention?

The author provides some comparison to other studies which look at, for example, environmental concerns and consumer purchase intention for home appliances, which is good. However, in the introduction the authors refer to several papers that explore consumers and insect food specifically. How do the results of this study compare with other studies that have specifically looked at insect food.

Table 4: I like how clearly this reports the results of the study. This is a great table.

Conclusions

P9 line 304: It is unclear what is meant by “businesses emphasize consumers product experience”.

The recommendations provided here to drive an increase in insect food purchase intention need to be stronger and to refer more clearly back to the results of this study.

Why should businesses push consumers to eat insects? The conclusion needs to emphasize why a business would want to do this, or what the benefit to consumers would be.

Author Response

 Please refer to the attached file (Annex 2)

Round 2

Reviewer 1 Report

Overall, the comments have been well addressed.

(1) I still have concerns about the wording “valid response rate” in the abstract. Although, clarification is provided in section 2.4. For the abstract, it would be more accurate to say something like “We distributed 408 surveys of which 77.45% were used in this study”

(2) In the introduction, my previous comment still applies: Lines 30-31: I’m not sure that SUPPLY is the main argument for eating insects – as much as other drivers such as climate change, land use, resource use, nutrition, etc. Consider reframing this issue given this information. You have address these issues very well in your revision, you might consider emphasizing these points, rather than production. As an introduction, consider saying something like “In light of the myriad environmental concerns we are faced with including global climate change…” since you have already added good information about GHGs and insects.

(3) This sentence requires rewriting for language and grammar. The information is great, but the grammar is not acceptable for publication the way it is: “Feed each kilogram of edible insect compared with blanket traditional livestock, only need 2 kg of feed on average, compare to keep a cattle to need 8 kg of feed to produce 1 kilogram of beef come of contributing to and cutting atmosphere scot, emission minor greenhouse gas, requirement of land and aqua also compare mammal and birds [2].”

(4) This is a good addition to the paper, however, it needs to be edited for language and grammar: “Past literature has found that insects have important effects in the nutritional history of Africa,

Asia and Latin America [4–6], Yukpa in Colombia and Venezuela Some insects are used in meat

dishes, and studies have also explored the zonocerus variegatus, bee larvae and pupae in southern

Nigeria with high protein [5,7]. Studies also have shown that insects are a good source of protein and micronutrients [5,6,8].”

(5) My previous comment still applies: Line 37-38: how insects-as-food is named is not the same as a definition, as you have pointed out. How are each of these names (insect food, insect-based food, insects as food) defined? If they are defined as eating insects, then the definition is not the issue. This is not clear in how you have phrased it. In other words, how do the definitions in these other studies differ from your definition, provided on line 43? Instead of “definition”, consider rewording to something like “no standard terminology”

(6) Excellent addition: “In remote rural and biodiversity-rich tropical countries, … In Taiwan, although it started late, there is an ecological rehabilitation of the Fuyang enterprises that have both green energy and alternative food.”

(7) The copy editor can confirm, but I believe when you use a direct quote, you need to provide a page number (for example, [9] (p. 3).)

Author Response

(The authors gave the same response as above.)
